# Illustrations of Coping and Mental Well-Being of Adolescents Living with HIV in Cape Town, South Africa During COVID: A Photovoice Study

**DOI:** 10.3390/ijerph21111517

**Published:** 2024-11-14

**Authors:** Yolanda Mayman, Talitha Crowley, Brian van Wyk

**Affiliations:** 1School of Public Health, University of the Western Cape, Bellville 7535, South Africa; bvanwyk@uwc.ac.za; 2School of Nursing, University of the Western Cape, Bellville 7535, South Africa; tcrowley@uwc.ac.za

**Keywords:** adolescents, adherence, antiretroviral therapy, COVID-19, HIV, mental well-being, psychosocial support, qualitative study

## Abstract

Adolescents living with HIV (ALHIV) are the fastest-growing population living with HIV globally. The COVID-19 pandemic disrupted health systems, thereby negatively affecting the quality and availability of HIV care and support services. This study describes the challenges and treatment experiences of ALHIV on ART at a public primary healthcare facility in a low-socioeconomic community in Cape Town, South Africa during the COVID-19 pandemic. A qualitative photovoice study was conducted with 21 adolescents (aged 14–19 years; 13 females and 8 males). Participants shared pictures illustrating their experiences during the COVID-19 pandemic in groups. Group discussions were audio-recorded and transcribed in full and subjected to content analysis. Seven themes emerged describing ALHIV’s personal, social, and economic challenges, their means of coping, and feelings of vulnerability. Challenges included the loss of significant others, lack of social support and opportunities, experiences of HIV-related stigma and discrimination in the household, loneliness, and isolation. In spite of difficulties, some ALHIV displayed resilience by continuing their medication routines during lockdown periods. After the COVID-19 pandemic, there is a need for the restoration of the health system and services, including psychosocial support to re-engage ALHIV in care and treatment.

## 1. Introduction

The 95-95-95 targets set by the United Nations Program (UNAIDS) for HIV/AIDS seek to end the AIDS epidemic by 2030. These targets call for 95% of people living with HIV (PLHIV) to know their HIV status, 95% to receive sustained antiretroviral therapy (ART) and 95% to achieve viral suppression by 2025 [1,2,3]. It is estimated that more than 8 million people are living with HIV in South Africa [4]. For the population group 15–49 years, an estimated 19.6% is living with HIV [5]. According to the South African Department of Health, more than 5.5 million people were registered to receive ART as of March 2023—making the ART program in South Africa the largest in the world [6,7]. However, optimal adherence to ART is crucial for ART success, which in turn increases the life expectancy of people living with HIV [8].

With improved antiretroviral regimens, the survival rates of children living with HIV improved, leading to increases in the prevalence of adolescents living with HIV (ALHIV). In 2022, there were an estimated 210,000 new HIV infections among adolescents and young adults aged 15–24 years in sub-Saharan Africa [9]. ALHIV (aged 10–19 years) are classified as a vulnerable population due to the unique biological, physical, and structural changes and challenges they often face during this developmental period of adolescence as they transition into adulthood [10]. This group experiences poorer outcomes at all stages of the HIV care continuum (uptake of testing, linkage to care, adherence, retention in care, and viral suppression) when compared to other groups living with HIV [11,12,13]. ALHIV face higher rates of psychological challenges compared to their peers without HIV [14]. Living with a chronic condition and experiencing concomitant HIV-related stigmatization while simultaneously dealing with normal adolescent development can be particularly harmful to the mental well-being of ALHIV. Most often, experiences of stigmas are negative in nature and can evoke feelings of shame, helplessness, and worthlessness within ALHIV [15,16]. Moreover, ALHIV often experiences internal and external stressors related to mental health conditions. Examples of internal stressors include feelings of isolation and shame related to their health and disclosure of their HIV status, which may lead to concerns over social acceptance. Examples of external stressors include social stigma related to HIV, which may lead to discrimination and bullying by peers, which in turn negatively affect their treatment adherence [17].

The outbreak of the COVID-19 pandemic disrupted economic, social, and health systems worldwide, including South Africa. The first confirmed COVID-19 case in South Africa was announced by Health Minister Dr Zweli Mkhize on 5 March 2020. On 15 March 2020, President Cyril Ramaphosa declared a national state of disaster, which introduced school closures, travel bans, and restrictions on public gatherings [18]. COVID-19 alert level 5 was implemented on 26 March 2020 until 30 April 2020, with risk-adjusted level 5 being implemented for a month, after which alert level 4 was introduced.

In this period, public health facilities were severely pressured due to increases in COVID-19 admissions, which meant that “other” patients who did not require life-saving intervention were delayed in receiving timely care [19]. By April 2022, a total of 417,636 cases of COVID-19 were confirmed in Cape Town Metropole [20]. The COVID-19 pandemic and its subsequent lockdown measures also introduced significant disruptions to the daily lives of children and adolescents around the world and affected the treatment and adherence of ALHIV to ART [21]. The number of young people struggling with mental health, including anxiety and depression, has more than doubled since the start of the COVID-19 pandemic [22,23]. ALHIV were at an increased risk of anxiety, depression, and prolonged isolation during the COVID-19 pandemic as existing challenges were further exacerbated [15]. COVID-19 regulations also affected interventions aimed at enhancing ART adherence as well as community-based support [2]. Additional barriers to HIV care and treatment services included a shortage in the supply of ART medication, a reduction in the provision of HIV services, and a shortage of resources [24,25].

Studying the experiences of ALHIV during the COVID-19 pandemic is essential for shaping effective public health policies. Currently, there is a dearth of research on the mid-to-long-term impacts of the pandemic on health outcomes in children and ALHIV, particularly in low- and middle-income countries (LMIC) [2,26,27]. Furthermore, it poses questions regarding the impact of the COVID-19 pandemic on the health and mental well-being of ALHIV, as research states that understanding the long-term effects of the pandemic on mental health and well-being is crucial for developing effective interventions that promote resilience and holistic well-being in children and adolescents [28]. Existing research on the impact of COVID-19 on PLHIV in sub-Saharan Africa has focused on clinical issues like coinfection and interruptions to care [14]. There is a need to garner a more comprehensive understanding of the impact of the COVID-19 pandemic on the health and well-being of ALHIV, especially in contexts of the hyper HIV epidemics in already disadvantaged communities with existing health system challenges. Understanding these long-term impacts can inform the development of targeted interventions and health system improvements to foster resilience and holistic well-being in children and adolescents in LMIC. These may, in turn, inform the formulation of health policies and regulations that better address the needs of ALHIV and ensure that future health responses to emergencies are inclusive and effective, while creating readiness within health systems to deal “better” with future pandemics. This paper describes the challenges and treatment experiences of ALHIV on ART in a low socio-economic setting in the Cape Town Metro in South Africa during the COVID-19 pandemic.

## 2. Materials and Methods

### 2.1. Study Design

Photovoice methodology was used in this study to explore the challenges and treatment experiences of ALHIV on ART during the COVID-19 pandemic. Photovoice is a participatory research design used to engage community members in research and allow them to share their lived experiences through photographs in an effort to promote dialog and initiate change [29,30,31]. Photovoice methodology has been applied in qualitative health-related research involving a range of topics, including barriers to healthcare and mental health [32]. Further, photovoice methods allow for the sharing of multiple truths and lived realities among participants and an increased understanding of the individual experiences of vulnerable groups such as children in research [33].

### 2.2. Study Context

The City of Cape Town is a largely urban area with a high population density [34]. The selected public health facility has been in operation since 2021 and offers services to a population of more than 200,000 residents in low socioeconomic income areas [35]. This facility seeks to improve the quality of life of children, adolescents, families, youth, and women through a range of interlinked activities, in partnership with local citizens, community-based organizations, non-governmental organizations, a range of strategic public and private partnerships, and the support of local and global donors. The health facility organizes care for patients with chronic conditions (such as HIV, diabetes, and hypertension) through chronic care clubs [35]. These clubs are regular meetings of patients with similar conditions at intervals that correspond to their scheduled clinic appointments or pick-up dates for medication. At these clubs, which consist of between 20 and 30 members, relevant health education and support services are provided. For ALHIV, youth clubs are organized where they are scheduled to pick up their medication or attend clinic appointments with peers of their age who are on ART in an attempt to improve adherence and engagement in care [35].

### 2.3. Study Participants

Participants were recruited from a public primary healthcare facility in a low socioeconomic community that operated HIV adherence clubs for youth and ALHIV in the Cape Town Metropole in the Western Cape province of South Africa. The research team introduced the study to key staff at the facility, who assisted with the identification and recruitment of eligible study participants. The inclusion criteria for participants were: (1) must be living with HIV, (2) between the ages of 14–19 years at the time of the study (this age group is a growing population among all people living with HIV globally), (3) receiving ART at the facility, (4) have been disclosed to [regarding their HIV status], (5) can speak and understand English, (6) be willing to safely take part in the project activities (taking of photos with cellphones), and (7) be willing to attend follow-up sessions to discuss photos in a peer setting. Purposive sampling was performed to identify participants who met the inclusion criteria and ensure representation of gender and age. Recruitment took place over a period of two months and was dependent on the willingness of ALHIV at the facility to participate and the number of phones available in the study. A total of 21 ALHIV were invited to participate in the study. This sample was deemed adequate as recurring themes began to emerge from the data. Seven focus groups were conducted with the participants.

Of the 21 ALHIV who participated in this study, 13 were females and 8 were males (see summary Table 1 and full description in Appendix A).

### 2.4. Procedures

The researcher and research assistant received training in photovoice methodology from an experienced HIV researcher who applied photovoice methods in HIV-related research. Study participants attended three sessions. At the introductory session, the staff and research team met and explained the study to eligible participants. Those who agreed to participate in the study received information and consent forms (additional parental/guardian consent forms were issued to those under 18 years) in a language of their choice (English, Afrikaans, or isiXhosa). Participants returned the following week with signed consent forms, which indicated consent to participate in the study. In this second session, the researchers explained the study in more detail to all consenting participants. Each participant received a cellphone with camera capabilities and was instructed to take as many pictures as possible that portray their experiences and challenges as ALHIV during the COVID-19 pandemic. At the conclusion of this session, a meeting date was agreed upon (usually the following week) where participants would meet as a group and present and discuss their photos in turn.

### 2.5. Data Collection

During the third and final session, participants returned to the facility as a group with the camera phones and were given the opportunity to share their photos. Photos were uploaded to a laptop and displayed through a mobile projector. Group discussions occurred after participants shared their pictures with other group members. Each session was facilitated by two members of the research team. Each focus group consisted of 3–6 participants per group and included male and female participants. Each photovoice session lasted 30–60 min and was digitally recorded and transcribed in full by the first author (YM).

### 2.6. Data Analysis

Participants’ photos were inserted into the discussion transcripts, where they were referred to during the group discussion session. All transcripts were prepared and uploaded to Atlast.ti and subjected to inductive content analysis (ICA). ICA is a method designed to identify and interpret the meaning of forms of data by isolating and organizing small pieces of data in a manner to describe or explain a phenomenon [36]. The five steps outlined for ICA include reading transcripts and familiarization with the data, first-round coding to organize the data and identify sections according to content, second-round coding to develop sub-themes and refine codes, refining and comparing sub-themes, and lastly, synthesis and interpretation of themes to create a narrative that gives an overall explanation of the phenomenon [37]. This analytic method allowed the description of the challenges and experiences of participants on HIV treatment during the COVID-19 pandemic as well as their mental well-being during this time.

### 2.7. Ensuring the Trustworthiness of This Study

The criteria developed to ensure trustworthiness and rigor in this study include credibility, dependability, confirmability, and transferability [38,39,40,41]. Credibility was achieved through the researchers asking clarifying questions during group discussions and reflecting on their own experiences during the process through reflective notes and peer debriefing. To achieve transferability, the research team provided an account of where and how the research was conducted to ensure that this research can be applied to other scenarios or groups. Dependability was achieved through the confirmation and verification of decisions made during each stage of the research process. Lastly, confirmability was achieved through journaling and the explanation of how findings were established.

### 2.8. Ethical Considerations

Ethics clearance for this study was obtained from the University of the Western Cape Biomedical Research Ethics Committee (BM23/3/7). The researchers sought verbal and written consent from all included study participants. All participants were also required to consent to keeping the information shared in groups confidential. Participants consented to the digital recording of discussions and the publishing of research findings. Parental consent was obtained for participants under the age of 18 years. Pseudonyms were used to protect the identity of participants and maintain anonymity. Participants were also instructed not to include family, friends, or any other individuals in pictures. Following the final group discussions, the researcher shared the contact information of counseling service centers with those who may have been emotionally affected. In the case where participants reported concerning information, permission was obtained to share these concerns with the staff at the facility. Accounts of previous suicidality among some participants were noted and reported to the health workers.

## 3. Results

### 3.1. Description of Themes

Seven main themes emerged from the focus group discussions with the ALHIV involved in this study: “Personal, social and economic impact of COVID-19”, “Leisure activities as a means of coping during the pandemic”, “Strategies to mitigate COVID-19 risk and vulnerability”, “Psychosocial consequences of COVID-19”, “Sources of support during COVID-19”, “Adherence to HIV treatment” and “Readjustment to the school environment after lockdown” (see Table 2). These are further discussed below.

### 3.2. Theme 1: The Personal, Social, and Economic Impact of COVID-19 Restrictions

The COVID-19 pandemic introduced restrictive measures and lockdown regulations meant to curb the spread of the coronavirus. These COVID-19 restrictions had various adverse effects on personal, social, and economic aspects of the lives of ALHIV. Many ALHIV experienced personal loss, the loss of opportunities, a lack of socialization, food insecurity, and various challenges within their homes that impacted their personal well-being.

#### 3.2.1. Personal Impact

The COVID-19 pandemic brought about a great deal of personal loss for the youth and ALHIV involved in this photovoice study. Participants experienced the loss of close family and friends as a result of COVID-19 and various other health-related issues. COVID-19 lockdown and movement restrictions meant that families were unable to visit loved ones in hospitals and attend the funerals of family members and friends. ALHIV in this study expressed their experiences of losing loved ones during the COVID-19 pandemic by reflecting on their sense of loss and grief.

“*I was sad in this picture because my father died so I was thinking my mother is supporting so much.*” (Participant 6, Female)

“*This one reminds me of my friend that passed away. He used to visit me during the lockdown. We enjoyed making short films and posting them on YouTube just to have fun and spend time. I can’t remember when he died but he died in a car accident.*” (Participant 9, Male)

Along with experiencing personal losses, COVID-19 restrictions took away numerous opportunities for ALHIV in this study. Participants reflected on losing academic scholarships and sporting opportunities due to the commencement of the lockdown. Activities such as birthdays and family vacations could also not be celebrated and enjoyed due to the imposed COVID-19 restrictions.

“*This is where I would usually look outside. My birthday is in July so I would love to go to Table Mountain. So I couldn’t go [there] on my birthday*” (Participant 17, Male) (Figure 1)


*“COVID affected me. I was playing rugby, and I discovered and scouted by the Western Province squad. All of a sudden COVID-19 came and I’m just very disappointed.”*
(Participant 16, Male)

#### 3.2.2. Social Impact

ALHIV often experienced feelings of boredom and loneliness because of the lack of socialization during lockdown periods. Many were not able to go outside and socialize with friends in the neighborhood. Additionally, during the COVID-19 pandemic, ALHIV missed engaging in social activities such as playing sports outside with friends.

“*Yeah because I missed my friends a lot, but I couldn’t be near them.*” (Participant 1, Female)

One female participant took a picture of her street (Figure 2) and described how she missed the social activities in her neighborhood:


*“So when I was bored staying inside the house I would just go stand in front of the house and look around the street. It was quiet at that time so maybe I’ll take a chair and sit there. There was no one. I missed playing soccer. When you go out there… there’s like a field where we play there. So, I just sat and missed playing soccer.”*
(Participant 4, Female) (Figure 2)

#### 3.2.3. Economic Impact

ALHIV in this study experienced food and electricity shortages within their homes. COVID-19 restrictions resulted in the loss of many jobs and a loss of income. In this study, the families of ALHIV were affected by the loss of income and employment as they were unable to purchase food due to price increases. Additionally, they were unable to cook food in their homes due to a shortage of money to buy electricity. Many households in lower-income communities resorted to using gas stoves for cooking purposes. However, this was not always possible due to the price of paraffin and gas.


*“So during COVID this 2 L cooking oil cost R100 I think. So it was very difficult for others at my home. We had to use a small oil when we were cooking because we were saving it. We cannot afford another one again and again.”*
(Participant 14, Female)


*“During COVID there was load shedding so at my home we used to cook with a gas stove if its load shedding. So it wasn’t easy for us because we were used to electricity. It’s not easy using this thing. You have to learn about it?”*
(Participant 14, Female) (Figure 3)

The COVID-19 pandemic sparked fears of an economic crisis and system collapse. However, measures implemented to combat the pandemic, such as social distancing, feelings of loneliness, and travel restrictions, resulted in the loss of employment, loss of income within households, and increased food insecurity and electricity shortages. Closed schools also meant that ALHIV experienced a lack of socialization, resulting in increased feelings of loneliness as a result of COVID-19 restrictions.

### 3.3. Theme 2: Leisure Activities as a Means of Coping During the Pandemic

During the COVID-19 pandemic, there were various activities that adolescents engaged in to keep themselves occupied during the pandemic and to socialize with those around them and in their households. Leisure activities were either solo or group activities.

#### 3.3.1. Solo Activities

ALHIV engaged in solo activities such as sleeping, listening to music, journaling and exercising. These activities not only kept them occupied but also kept their minds off the reality of the pandemic.


*“This pool board, it’s kind of like my best friend, but I used to play all the time with him when I got a little lonely or sad. It makes me happy and when I feel a little down it makes me happy.”*
(Participant 8, Male) (Figure 4)

#### 3.3.2. Group Activities

Group activities often included playing games and listening to music with other friends and family members within the home.


*“During COVID we had to find things to do because it was boring at the house. So I got games to play with the children and big people because they also got bored.”*
(Participant 4, Male)


*“So me and my cousin used to play FIFA to forget that we were inside. Yeah we also used to play 30 s, Monopoly.”*
(Participant 15, Male)

Leisure activities proved to be effective in keeping adolescents and youth occupied during the pandemic. Additionally, the restrictions facilitated family bonding as more group activities took place within the home.

### 3.4. Theme 3: Strategies to Mitigate COVID-19 Risk and Vulnerability

Vulnerable groups such as the elderly and individuals with chronic illnesses were deemed more at-risk of contracting COVID-19. To mitigate this risk, ALHIV engaged in activities such as cleaning to increase the hygiene within their homes. Participants in this study also chose to limit contact with individuals outside of their homes in an attempt to decrease their risk of COVID-19 infection.

#### 3.4.1. Increased Hygiene

Some ALHIV in this study expressed that they felt vulnerable to catching COVID-19 because of their HIV status. Participants felt that their status put them at an increased risk of contracting the coronavirus. To prevent this, many ALHIV increased their household chores and hygiene within their homes to decrease the chances of them catching COVID-19.


*“I chose to do chores, overdoing chores. Even if my sister sits down I’m wiping where she was sitting.”*
(Participant 19, Female)

#### 3.4.2. Self-Isolation

Another way in which ALHIV tried to prevent catching COVID-19 was to self-isolate within their homes. Many participants chose to stay indoors to prevent themselves from coming into contact with individuals outside who may have COVID-19, thus placing them at risk of also catching the virus.


*“Because I didn’t like being between other people because I would get COVID from them. I was much safer in my home.”*
(Participant 14, Female)

The risk of contracting COVID-19 resulted in participants in this study employing various strategies to protect themselves from exposure to the virus. This shows that adolescents and youth with HIV are aware of their vulnerability due to their HIV status and actively engaged in behaviors to not expose themselves to the coronavirus or individuals who may have had it.

### 3.5. Theme 4: Psychosocial Consequences of COVID-19

COVID-19 restrictions interrupted the daily routines of ALHIV, which negatively affected their psychosocial development and outcomes. Many ALHIV in this study experienced loneliness and conflict within their homes. These factors triggered emotional responses within ALHIV, with some expressing instances of suicidality as a means of wanting an escape.

#### 3.5.1. Social Isolation

ALHIV felt isolated within their homes during the COVID-19 pandemic. Many participants in this study experienced frustration and loneliness as they were not able to see and interact with their friends. Others experienced feelings of being caged and trapped inside their homes, often leaving them with a desire to escape.


*“So I felt like I’m all alone and the gates are closing. So I felt like let me just help them close cause I’m all alone and by myself.”*
(Participant 3, Female) (Figure 5)


*“There were a lot of fights with my sister, and I wanted to burn the house.”*
(Participant 3, Female)

#### 3.5.2. Household Conflict

In conjunction with experiencing feelings of loneliness, many ALHIVs in this study witnessed conflict within their homes. These experiences often resulted in adolescents having to relocate to neighbors or other family members to escape the conflict and ensure their own safety. In some instances, familial conflict within the home also negatively affected the HIV treatment adherence of ALHIV due to the temporary relocations to places of safety.


*“So this picture I would just go at the back of the house, and I’d just stay there and cry. During the day they would argue so I just go back and cry and even pray sometimes. I didn’t know how to pray so I would say just get me out of here. So I’d cry a lot. My brother was like you don’t need to cry. We’re going to go stay with my aunt. I didn’t want to leave cause I was scared that where I went it would be worse cause my aunt also stayed with her boyfriend. So I was like ‘No I’ll stay here’. But then I moved eventually.”*
(Participant 6, Female)

#### 3.5.3. Emotional Responses

Some participants shared stories of their suicide attempts during the COVID-19 pandemic. Causal factors behind feelings of suicidality often stemmed from conflict and targeted abuse within the home. Participants in this study also expressed feelings of sadness and loneliness, which added to thoughts of suicidality, with one participant being hospitalized as a result of a suicide attempt.


*“I did. It’s when I took my tablets and thought I could overdose and kill myself. That’s how I felt but I couldn’t do it because there were people. Every time I try there is someone who catches me and says I can’t do this. This is not right. Then I’m like okay I’ll get past this. Then she starts again and I’m like I should’ve killed myself long ago if it wasn’t for that person. Then another person comes in and says no don’t do that. Then I leave it.”*
(Participant 5, Female)

Experiences of stigma due to HIV status emerged during focus group discussions with ALHIV. ALHIV experienced negative treatment in the form of targeted bullying by community and family members. As a result of this harassment, many ALHIV feared that their HIV status would be disclosed to members of the community during the pandemic. Participants in this study shared that family members disclosed their HIV statuses, resulting in them feeling dirty and like an outcast.


*“I wanted to kill myself because I felt like my sister doesn’t like me. So I felt like no let me kill myself so that she won’t talk about me if I have HIV.”*
(Participant 5, Female)


*“Yes, because she was saying because she was telling people that I’m HIV positive…Okay at first during the COVID when it started, everybody called me bad names. My friends, my sisters friends. I felt like trash or a dustbin where you can throw your dirty things and just leave it there. That’s how I felt, and no one was there to clean me or to help me come clean with myself and be happy again. So, that’s how I felt during COVID.”*
(Participant 3, Female) (Figure 6)

While the majority of ALHIV in this study expressed experiencing negative emotions in response to the COVID-19 pandemic, some participants experienced feelings of optimism and self-motivation, which helped them remain positive during the pandemic.


*“This picture [shows picture of him smiling] is telling me that I always have a smile on my face. No bad stuff. Even when I had corona, nothing stopped me. Nothing will take my smile off or make me feel bad. Always have a smile on my face.”*
(Participant 11, Male)

The above findings highlight the impact of the pandemic on the psychosocial well-being of adolescents and youth with HIV. Feelings of loneliness as well as experiences of conflict and discrimination had adverse effects on ALHIV, as two participants expressed instances of suicide attempts. These results also show the lasting effects of the COVID-19 pandemic.

### 3.6. Theme 5: Sources of Support During the COVID-19 Pandemic

Participants highlighted various sources of support within their immediate environment and community, which helped them process their emotions and experiences during the pandemic. Social and spiritual sources of support, such as family and friends, as well as spiritual leaders, were instrumental in cases where participants felt anxious and uncertain.

#### 3.6.1. Social Support

Family members and friends were common sources of support for ALHIV during the COVID-19 pandemic, as participants often shared their feelings and concerns with family and friends.


*“When COVID-19 started me, and my friends were always together playing. I didn’t know. Some of them told me coronavirus was in, so I was shocked because I didn’t know. So they told me not to worry. They will always be beside me no matter what because it was like corona, corona, corona. So they cooled me down and played with me. So they made me smile.”*
(Participant 8, Male)


*“My family was there. They were very supportive.”*
(Participant 9, Male)

Another source of support for ALHIV and their families during the pandemic was the support provided by the government. Individuals in low socioeconomic communities received food parcels, which participants say were a great help during the pandemic.


*“Okay, here this is a cupboard. As a family we didn’t have enough time to go to the supermarket to buy food, so we were dependent on the food parcels that were supplied by the government. There was a truck on the road.”*
(Participant 9, Male) (Figure 7)

#### 3.6.2. Spiritual Faith and Guidance

ALHIV in this study sought spiritual guidance through their faith, prayers, and by consulting with traditional healers during periods of conflict and uncertainty during the COVID-19 pandemic.


*“So me and my brother went to this place. Sangoma to cleanse out what was happening in our home. We thought it wasn’t normal cause it was that bad. So then they told us your mother is supposed to leave this man. You two are supposed to be praying for your mother, praying for your mother because there are more things coming. The only strong structure is you guys. You need to protect yourself and consult with your ancestors and pray for protection. So this represents that.”*
(Participant 6, Female)

Support was a positive factor during the COVID-19 pandemic for participants in this study. Family and friends allowed ALHIV to express their thoughts and concerns during the pandemic, while faith and spiritual support offered guidance to participants in this study.

### 3.7. Theme 6: Adherence to HIV Treatment

Adherence to HIV treatment during the COVID-19 pandemic was influenced by various facilitators and barriers, which either led to adherence or non-adherence to HIV treatment.

#### 3.7.1. Facilitators

Facilitators such as self-agency allowed ALHIV to remain adherent to their treatment. The collection and delivery of HIV medication exemplify the personal discipline among ALHIV that helped maintain adherence to HIV treatment.


*“My grandparents used to go fetch them for me.”*
(Participant 13, Female)


*“Like Dr [name] has made it easy for us. Like, we didn’t have to like to be in one place. So like people were collecting their medication and leave. So we didn’t have to stay there.”*
(Participant 1, Female)

#### 3.7.2. Barriers

Barriers refer to internal or external factors that hinder adherence to HIV treatment amongs adolescents. Barriers to treatment included conflict within the home, resulting in ALHIV having to flee to the homes of neighbors and family members for safety. Disrupted routines during the pandemic also meant that participants often forgot to take their HIV medication.


*“Usually I used to forget sometimes but I did take my medications. I used to drink medications at 9 most days so I would forget sometimes or wake up late.”*
(Participant 17, Male)


*“Yeah. Sometimes maybe when it’s late then I didn’t take my medication. Then when he’s back he will start shouting and stuff. If he’s my stepfather I’m mostly related to my mother. What happens to her also affects me. I can even forget about medication and focus on this matter.”*
(Participant 20, Male)

HIV treatment adherence was important to ALHIV in this study, as many displayed self-agency to ensure that they took their medication. Disrupted routines caused by conflict in the home highlight the impact of external factors on the treatment adherence of ALHIV.

### 3.8. Theme 7: Readjustment to the School Environment After Lockdown

The introduction of the reopening of schools meant that ALHIV had to readjust to the school environment. Many ALHIV engaged in homeschooling activities that involved them having to teach themselves and stay up to date with their academics. Participants expressed that this created a sense of academic independence, which motivated them to do their schoolwork during the pandemic. However, while many expressed feelings of excitement to return to school, some were not as optimistic and stated that they preferred learning on their own.

#### 3.8.1. Academic Independence

Participants in this study expressed that they experienced academic independence because they were forced to take responsibility and complete their schoolwork without the help of educators and study groups.


*“During the COVID-19 children couldn’t go to schools because schools are closed for a very long time and that led to homeschooling. People were applying to online schools during COVID-19 because if they would just sit at home and do nothing that would affect their results at the end of the year because teachers would just continue and continue with term four’s work and you guys would skip term three’s work. Just tell you just study everything where we left off and the stuff that we didn’t do during the COVID-19. Children who are really dedicated to work would do schoolwork on a daily basis just to keep them updated so at the end of the year they found themselves to be good and that led to homeschooling. Some of the children are not going to campus school. They are going to homeschools schooling just because of COVID-19.”*
(Participant 16, Male)


*“I was able to concentrate because I was on my own doing my schoolwork.”*
(Participant 13, Female)

#### 3.8.2. Feelings of Academic Isolation

In contrast, ALHIV often felt anxious and lonely during the pandemic because of the loss of their academic support and study groups.


*“So this picture [shows picture of her sitting with her schoolbooks], like it was me like studying alone. Since like we couldn’t be like a group of girls studying together and all that stuff. That really cost me a lot cause like I don’t know how to study on my own.”*
(Participant 1, Female)


*“And during that time I was smart. I was good. Always in the top five, top three ma’am. When COVID-19 came I had to stay indoors. Didn’t get a chance to go out. Always sleeping, I’ve been feeling lazy, and a lot happened.”*
(Participant 16, Male) (Figure 8)

The closing of schools during the pandemic meant that home and online schooling became the norm for adolescents and youth. While many were able to take responsibility for their studies, some participants expressed that the change in learning adversely affected them. This highlights the direct impact of the pandemic on the mental and academic state of ALHIV.

## 4. Discussion

The aim of this study was to describe the challenges and treatment experiences of ALHIV on ART treatment during the COVID-19 pandemic. Our study findings confirm that the COVID-19 pandemic’s impact on ALHIV is multifaceted—encompassing both direct and indirect effects on their health and mental well-being. Previous research demonstrated that social restrictions to mitigate the spread of SARS-CoV-2 led to the diversion of essential resources, reduced economic activity, and heightened food insecurity within households [42]. Furthermore, the pandemic disrupted healthcare delivery and presented significant challenges to HIV treatment programs, thereby exacerbating existing challenges within healthcare facilities and adversely affecting HIV care and service delivery to adolescents [2]. It was noted that the lockdown restrictions hindered access to clinical care and social support while exacerbating the mental health challenges of ALHIV [43,44].

Our study highlights that COVID-19 negatively affected ALHIV on personal, social, and economic levels. The loss of loved ones and the loss of academic opportunities unfavorably impacted participants’ mental well-being. Existing literature suggests that pandemic-related strategies have contributed to the deterioration of mental health among adolescents [15,45]. Many young people were exposed to various stressors during the pandemic, leading to a decline in their quality of life [46,47,48]. Similarly, social disruptions following severe restrictive measures had a cascading effect on youth and adolescents. A lack of socialization with peers at school and within communities often left ALHIV in this study with feelings of boredom and loneliness. School closures and social gathering restrictions intensified social isolation and feelings of confinement within the home among adolescents and youth [49,50]. The COVID-19 pandemic and lockdown regulations resulted in job losses, reduced income, and reduced food availability [51]. Our study found that food insecurity and electricity shortages were common in South African households as families experienced job losses and the loss of income as a result of lockdown restrictions. Previous research also shows that youth reported increases in the cost of living, food scarcity, and reduced income due to COVID-19 restrictions [12].

Adolescents in this study adopted various solo and group activities such as games and exercise to cope and remain active during the pandemic. A 2020 survey revealed that 64% of adolescents aged 10 to 18 years reported that their outdoor activity participation decreased during the early months of the COVID-19 pandemic [52]. During lockdown periods, many adolescents explored their potential by spending time typically used to engage with others to engage in personal interests [53]. These findings align with our study, as many adolescents engaged in creative activities such as journaling and listening to music with others as forms of self-expression.

During the COVID-19 pandemic, national governments worldwide implemented measures and strategies in an effort to curb the viral transmission of the coronavirus. Concentrated efforts included widespread closures of public places, travel restrictions, mandatory mask-wearing, as well as ‘stay-at-home’ orders [54]. In our study, ALHIV followed the implemented pandemic measures by increasing their personal hygiene and hygiene within their homes. Furthermore, ALHIV in this study felt that isolating and limiting contact with individuals outside kept them safe and decreased their risk of COVID-19 infection.

ALHIV are a vulnerable population, with higher loss to follow-up and adverse treatment outcomes, which placed them at an increased risk during the pandemic [12]. The psychosocial consequences of the pandemic included feelings of loneliness, household conflict, and severe emotional responses. Evidence suggests that in South Africa, vulnerability to domestic violence within households was further exacerbated during the COVID-19 lockdown [55]. Reasons for this include the confinement of women and children within the home with abusive and alcoholic perpetrators—which was confirmed in our study. Prolonged feelings of loneliness and confinement within the home were also linked to poor mental health outcomes during the pandemic, adding to an increased risk of experiencing violence within households [55].

Our study found that participants experienced suicide attempts and HIV-related stigma during the pandemic as a result of their HIV status. The disruption of protective factors such as school and social support for adolescents and youth during the pandemic adversely affected the mental health of this group. In a study conducted in the United States of America, it was found that the number of youths who died by suicide during the first 10 months of the pandemic was higher than the expected number of deaths had the pandemic not occurred [56]. This highlights how the pandemic increased the risk of suicide among youth and adolescents during lockdown periods.

Health emergencies such as the COVID-19 pandemic can have long-lasting psychological and physical consequences for ALHIV. Individuals with HIV may experience a more severe psychosocial impact, as a lack of interaction with others may lead to the disintegration of social bonds and a loss of support [57]. Family, friends, and spirituality/religion were essential sources of support for participants in our study. It has been reported that spirituality predicted well-being during the pandemic [58]. Another study found that the utilization of religious and spiritual beliefs led to higher levels of hopefulness and lower levels of fear and worry during the pandemic [59]. Participants in this study sought spiritual guidance and held onto their spiritual and religious beliefs during periods of uncertainty. The aforementioned evidence therefore confirms that social and spiritual sources of support positively impacted the well-being of ALHIV during the pandemic.

There are facilitators and barriers that influence adherence to HIV treatment. Facilitators such as self-agency and personal discipline motivated ALHIV in this study to collect their HIV medication and remain adherent during the COVID-19 pandemic. In contrast, barriers disrupted the routines of participants in this study, often resulting in them forgetting to take their medication due to household conflict and changes in living arrangements. A 2021 study found that the ART adherence of ALHIV was negatively affected due to them forgetting to take their antiretroviral drugs (ARVs) as well as changes in their schedules due to their living situations [60].

Apart from the home and family environment, the school environment has been recognized as crucial for the psychological well-being and recovery of young people [61]. Due to COVID-19 restrictions, conventional schooling was disrupted, and alternative modes of schooling such as homeschooling and online learning were introduced to maintain the academic routines of students. This required students to adjust to the new learning modalities and cultivate a sense of academic independence, which many ALHIV in this study managed to navigate. Research indicates that learners often experienced difficulties accessing online material, and many did not have quiet places to study during the pandemic [62]. The closure of schools also resulted in the loss of academic support, often leading to feelings of academic isolation. It has also been suggested that the closure of schools, diminished social engagement, and lack of academic support negatively affected the academic performance and confidence of students [61]. We recommend that the psychological recovery and resilience of students be prioritized to re-establish a sense of student connectedness and continued academic confidence.

## 5. Study Limitations

This qualitative study has some limitations. The sample was limited to those who attended and received ART at the facility; thus, our findings may not be applicable to those ALHIV who are not in youth clubs, as the latter may have different experiences of living with HIV and dealing with their chronic condition. Further, our findings are contextual and thus specific to this setting, limiting their wider application to the rest of the Cape Town Metro, province, and country.

## 6. Conclusions

Our findings highlight the coping strategies and mental well-being of ALHIV on ART during the COVID-19 pandemic. The restrictive measures aimed at controlling the outbreak introduced new challenges and exacerbated existing difficulties in various aspects of their lives. Feelings of loneliness and a lack of interaction with peers resulted in mental health challenges amongst ALHIV. Results also show that increased time indoors led to adolescents experiencing conflict within the home and instances of HIV-related stigma. To address these issues, we recommend prioritizing the mental well-being of ALHIV in the development of psychosocial and adherence support interventions. Such initiatives can enhance treatment outcomes and provide valuable insights for navigating future health crises, ensuring that ALHIV continue receiving essential treatment and support.

An effective approach to supporting adolescents during health crises could involve the creation of virtual youth support groups, enabling access to continuous mental and psychosocial support. Such groups would help mitigate negative psychological experiences and foster a sense of community among adolescents facing similar challenges, i.e., psychosocial support from peers. A pertinent example of this approach is the mobile phone-based intervention InTSHA (Interactive Transition Support for Adolescents with HIV), which incorporates closed group chats that facilitate peer support and enhance communication between ALHIV, their caregivers, and healthcare providers [63]. We recommend more exploratory qualitative studies with ALHIV in other settings to provide comparative insights into their experiences and needs, which would inform theory on how ALHIV are affected by health crises like COVID-19, and advance interventions and develop programs for ALHIV on ART in South Africa and the rest of the sub-Saharan African continent.

## Figures and Tables

**Figure 1 ijerph-21-01517-f001:**
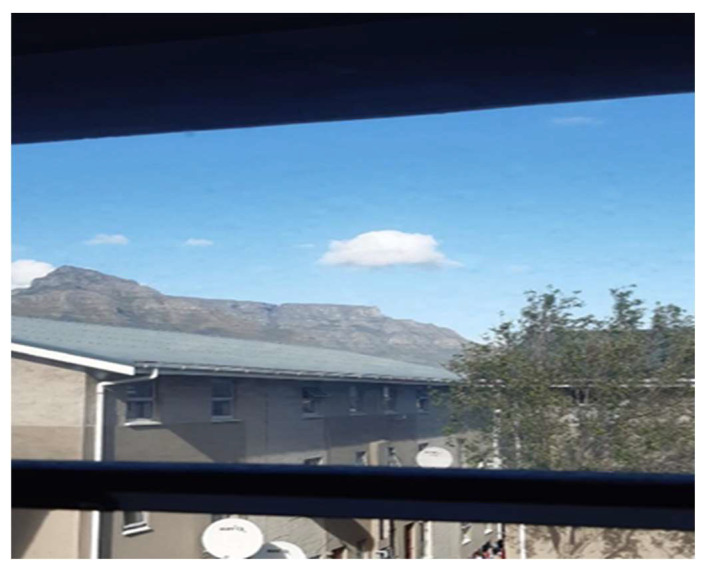
Missed being outside.

**Figure 2 ijerph-21-01517-f002:**
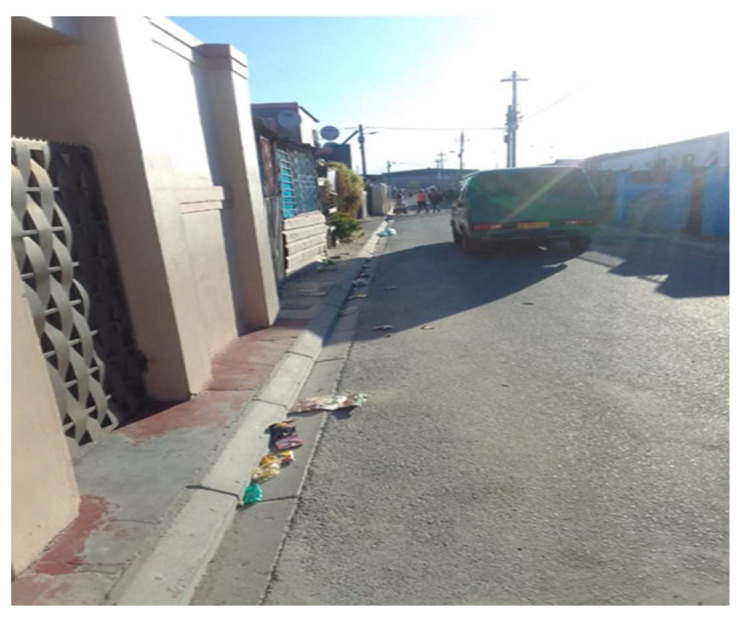
Quiet street.

**Figure 3 ijerph-21-01517-f003:**
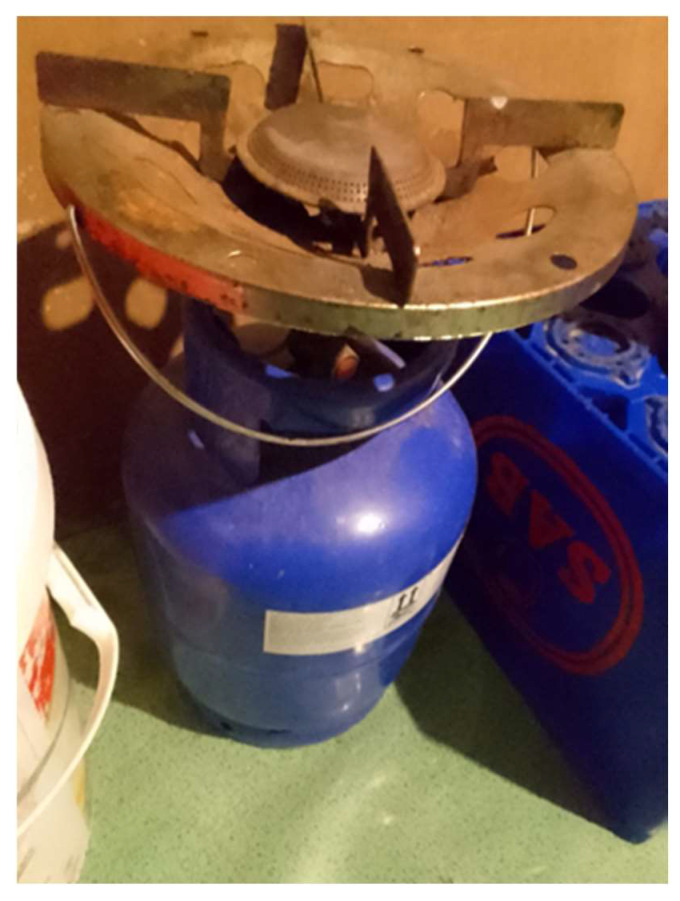
No electricity in the house.

**Figure 4 ijerph-21-01517-f004:**
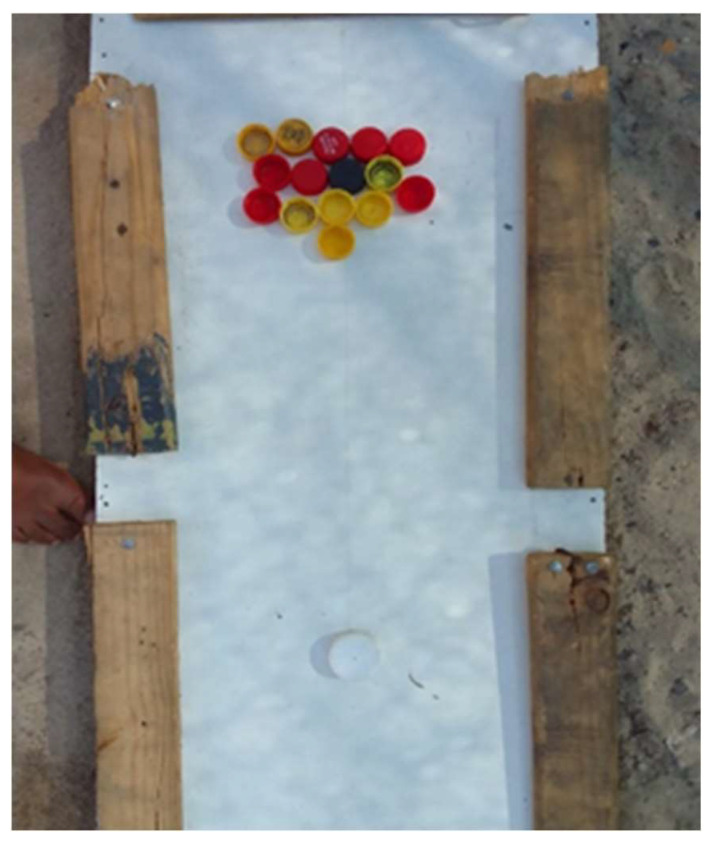
Playing indoor board games.

**Figure 5 ijerph-21-01517-f005:**
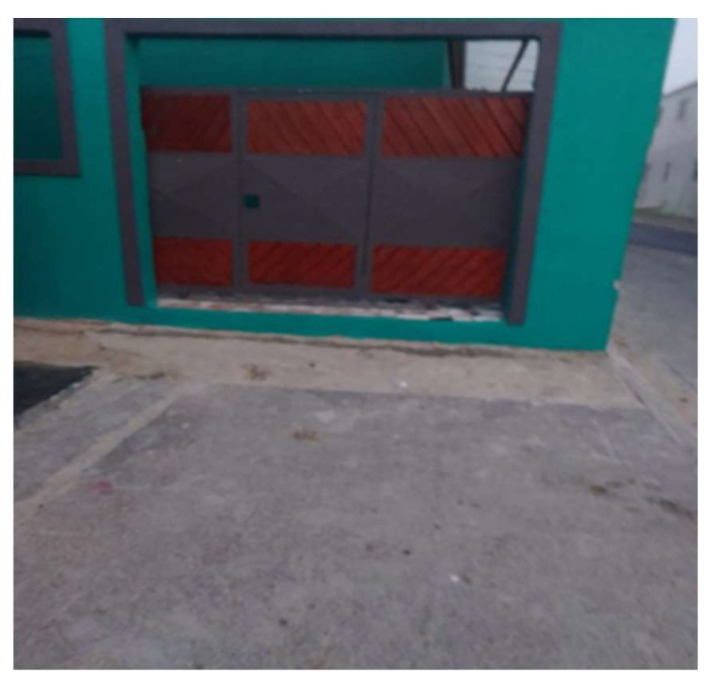
Locked in.

**Figure 6 ijerph-21-01517-f006:**
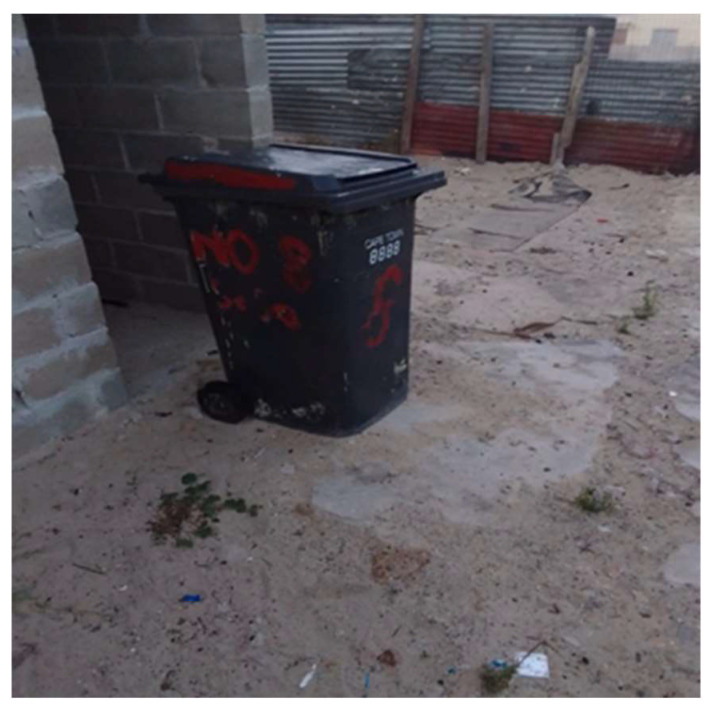
“I felt like trash”.

**Figure 7 ijerph-21-01517-f007:**
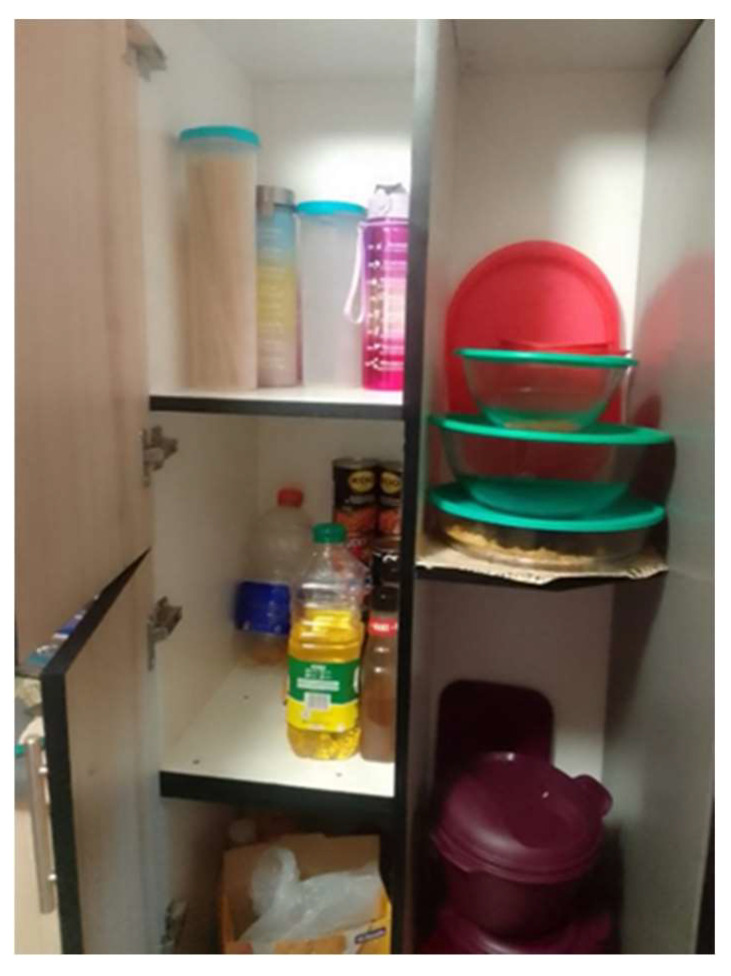
No food in our cupboards.

**Figure 8 ijerph-21-01517-f008:**
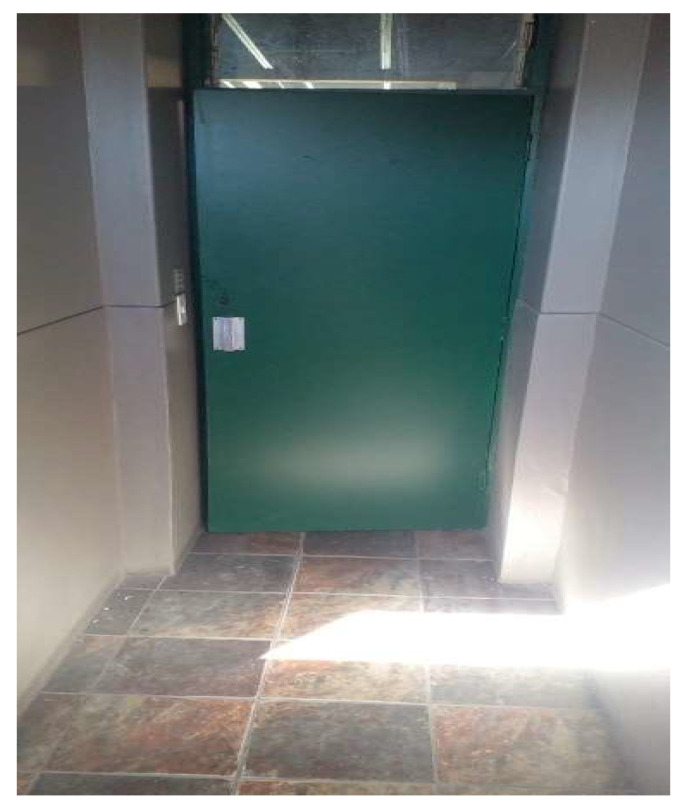
Staying indoors.

**Table 1 ijerph-21-01517-t001:** Summary description of participants (N = 21).

Category	Sub-Category	Total
Age	14–16	5
	17–19	16
Sex	Male	8
	Female	13

**Table 2 ijerph-21-01517-t002:** Description of themes, sub-themes, and codes.

Theme	Sub-Theme	Code
Personal, social, and economic impact of COVID-19	Personal impact	Loss of close family and friends
		Missed opportunities and social activities
	Social impact	Lack of socialization
	Economic impact	Food insecurity
		Loss of income
		Shortage of electricity
Coping through leisure activities	Solo activities	Listening to music
		Journalling
		Exercising
	Group activities	Playing games
		Listening to music with others
Strategies to mitigate COVID-19 risk	Increased hygiene	Increased household chores and cleanliness
	Self-isolation within the home	Decreased outside contact with other individuals
Psychological and social consequences of COVID-19	Social isolation	Feelings of sadness and loneliness
	Household conflict	Interpersonal and familial conflict
	Emotional responses	Suicidality
		Experiences of discrimination and HIV stigma
		Optimism
Sources of support	Social support	Family and friends
	Spiritual faith and guidance	Government food parcels
		Faith and traditional healers
Adherence to HIV treatment	Facilitators	Self-agency
		Delivery of medication
	Barriers	Disrupted routines
Readjustment to school after lockdown	Academic independence	Home school activities
	Feelings of academic isolation	Feelings of anxiety due to readjustment

## Data Availability

The data for this study are available upon request from the principal and corresponding author, Y.M.

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
