# Peer review of "Illustrations of Coping and Mental Well-Being of Adolescents Living with HIV in Cape Town, South Africa During COVID: A Photovoice Study"

_ijerph, 2024, doi:10.3390/ijerph21111517_

Round 1

Reviewer 1 Report

Comments and Suggestions for Authors

 Dear authors,

 My feeling is that your article is original and interesting. In this study I see that you examined the experiences of adolescents living with HIV under ART in a public primary healthcare facility in a low-income community during COVID-19 pandemic. In General, the article is well written and methodologically adequate. However, some points should be revised, to improve the relevance of the results. Below, I list some points with the aim of improving the quality of the study:

1.     The conclusion presented in the summary is generic - I suggest reformulation. What, in fact, does the analysis of the main results suggest?.

2.     Either in the keywords or in the title of the study, I suggest you add the expression: “qualitative study”.

3.     In the first paragraph, line 31, the authors state the 95%-95%-95% UNAIDS target is aimed until 2023, but this is incorrect. In 2023, international guidelines were proposed to control the HIV/AIDS epidemic by 2030. Their aim was to identify 95% of HIV cases, ensuring optimal treatment adherence for 95% of those identified, and ensuring that 95% of those living with HIV have an undetectable viral load.

4.     In line 40 of the introduction, a bibliographic reference is missing, as well as explaining to the reader whether the estimated cases are in the world, or on the African continent.

5.     Still in the second paragraph of the introduction, I suggest that the authors add the argument that HIV-related stigma can be especially harmful among adolescents because at this stage of psychic development the individual does not yet have the neuronal and behavioral maturity to develop coping mechanisms appropriate, and it is very possible that living with HIV at this stage, of personality and self-esteem formation, may produce problems associated with experiences of rejection and social isolation due to stigma.

6.     In line 47, the authors mention internal and external stressors. What would these stressors be? It would be relevant to give some examples.

7.     Countries presented very different responses to the COVID-19 pandemic. Therefore, it would be very relevant for the authors to present in the introduction an overview of what the pandemic was like in South Africa, in epidemiological terms, consequences, ways of coping, and the specificities of the health crisis in the city of CapeTown, where the participants live. Another point that needs to be addressed is how public health services were affected, in general.

8.     The justification for the study is quite generic. I suggest that the authors reformulate the justification thinking about the following question: how studying the experiences of adolescents living with HIV during the pandemic can be useful in practical terms for the formulation of public health policies? (keep in mind that you chose a Public Health Journal).

9.     As this is a qualitative study, with relatively few participants (n=21), it would be relevant to present in the results, or in the method (description of the participants) a table with the sociodemographic variables of each participant.

10.   I also consider it relevant to characterize the region in which the study was carried out and the health service selected.

11.  I suggest that the authors explain whether there were any criteria for selecting 21 participants, and not more or less. Was there any saturation criterion, such as theoretical saturation?

12.  The method also needs to include information on how many participants there were in the service (adolescents with HIV), how many were invited to participate, how many accepted, and how many participated in the three meetings.

13.  In the results, on line 326, page 9, the authors mention “feelings of depression”. However, technically, it would be better to talk about feelings of sadness and feelings of discouragement, since depression is not exactly a feeling. The same applies to the expression: “feelings of isolation”. My suggestion would be to change it to “feeling of loneliness”. I advise authors to change these expressions throughout the text (for example, line 358).

14.  In the discussion, in line 562, the authors make some recommendations, which I considered generic. I believe that the article would gain in originality and importance if the authors added a few paragraphs here with clear recommendations for caring for the well-being of adolescents with HIV during health crises, based on the results of this study.

I congratulate the authors on choosing such a relevant topic and I hope that the considerations presented will increase the quality of the study, so that it can be published. Best wishes to the authors.

Reviewer 2 Report

Comments and Suggestions for Authors

The paper is well written I suggest the following: First, you have collected data from 14 to 19 year old adolescents yet, you say your sample was from 10 to 19 years, please correct this. Second,  be consistent in your quotes.  Sometimes you indicate gender and age, in others you do not. Third, please indicate how many participants considered suicide in your results summary. 

Reviewer 3 Report

Comments and Suggestions for Authors

Thank you for the opportunity to review this insightful article on young people living with HIV during the COVID-19 pandemic. I have a few suggestions for the authors:

  • The Abstract is clear and concise, effectively summarizing the key points of the article.
  • The Introduction provides a well-rounded overview of the theoretical background. However, I would recommend adding a more explicit and well-argued presentation of the research questions toward the end of this section, especially considering the innovative methodology used.
  • In the Methods, it would be helpful to provide more detailed information about the sample. Interestingly, the Abstract includes more sample details than the Methods, which seems inconsistent.
  • Some sample data is currently presented in the Results section, which would be more appropriately placed in the Methods. The results themselves are clear and well-presented, and I appreciate the inclusion of citations and photos. Would it be possible to enlarge the photos in the final version of the article for better visibility?
  • The Discussion is clear and logically follows the results. However, I suggest including a brief note on the study's limitations.

Round 2

Reviewer 1 Report

Comments and Suggestions for Authors

The authors have excellently complied with all the requests for reformulation and clarifications that I made in my first review. Therefore, I consider that the article is fit to be published as it is.